# Iron Status of Infants in the First Year of Life in Northern Taiwan

**DOI:** 10.3390/nu12010139

**Published:** 2020-01-03

**Authors:** Chiao-Ming Chen, Shu-Ci Mu, Chun-Kuang Shih, Yi-Ling Chen, Li-Yi Tsai, Yung-Ting Kuo, In-Mei Cheong, Mei-Ling Chang, Yi-Chun Chen, Sing-Chung Li

**Affiliations:** 1Department of Food Science, Nutrition, and Nutraceutical Biotechnology, Shih Chien University, Taipei 10462, Taiwan; charming@g2.usc.edu.tw (C.-M.C.); mei11792004@gmail.com (I.-M.C.); mlchang@g2.usc.edu.tw (M.-L.C.); 2School of Medicine, Fu-Jen Catholic University, New Taipei City 24205, Taiwan; musc1006@gmail.com; 3Department of Pediatrics, Shin-Kong Wu Ho-Su Memorial Hospital, Taipei 11101, Taiwan; Ylchen1219@gmail.com (Y.-L.C.);; 4School of Nutrition and Health Sciences, College of Nutrition, Taipei Medical University, 250 Wu-Hsing Street, Taipei 11031, Taiwan; ckshih@tmu.edu.tw (C.-K.S.); yichun@tmu.edu.tw (Y.-C.C.); 5School of Medicine, Taipei Medical University, Taipei 11031, Taiwan; 6Institute of Environmental and Occupational Health Sciences, College of Public Health, National Taiwan University, Taipei 10617, Taiwan; 7Department of Pediatrics, Shuang Ho Hospital, Ministry of Health and Welfare, Taipei Medical University, New Taipei 23561, Taiwan; pedkuoyt@tmu.edu.tw; 8Department of Pediatrics, School of Medicine, College of Medicine, Taipei Medical University, Taipei 11031, Taiwan

**Keywords:** infant, breast milk, formula milk, iron deficiency, iron deficiency anemia

## Abstract

Iron deficiency (ID) and iron deficiency anemia (IDA) typically occur in developing countries. Notably, ID and IDA can affect an infant’s emotion, cognition, and development. Breast milk is considered the best food for infants. However, recent studies have indicated that breastfeeding for more than six months increases the risk of ID. This study investigated the prevalence of ID and IDA, as well as the association between feeding type and iron nutritional status in northern Taiwan. A cross-sectional study was conducted on infants who returned to the well-baby clinic for routine examination from October 2012 to January 2014. Overall, 509 infants aged 1–12 months completed the iron nutritional status analysis, anthropometric measurement, and dietary intake assessment, including milk and complementary foods. The results revealed that 49 (10%) and 21 (4%) infants in their first year of life had ID and IDA, respectively, based on the World Health Organization criteria. Breastfed infants had a higher prevalence rate of ID and IDA than mixed-fed and formula-fed infants (*p* < 0.001). Regarding biomarkers of iron status, plasma hemoglobin (Hb), ferritin, and transferrin saturation (%) levels were significantly lower in ID and IDA groups. The prevalence of ID and IDA were 3.7% and 2.7%, respectively, in infants under six months of age, but increased to 20.4% and 6.6%, respectively, in infants above six months of age. The healthy group had a higher total iron intake than ID and IDA groups, mainly derived from infant formula. The total dietary iron intake was positively correlated with infants’ Hb levels. Compared with formula-fed infants, the logistic regression revealed that the odds ratio for ID was 2.157 (95% confidence interval [CI]: 1.369–3.399) and that for IDA was 4.196 (95% CI: 1.780–9.887) among breastfed infants (*p* < 0.001) after adjusted for all confounding factors (including gestational week, birthweight, sex, body weight percentile, body length percentile, age of infants, mothers’ BMI, gestational weight gain, education level, and hemoglobin level before delivery). In conclusion, our results determined that breastfeeding was associated with an increased the prevalence of ID and/or IDA, especially in infants above six months. This suggests that mothers who prolonged breastfeed after six months could provide high-quality iron-rich foods to reduce the prevalence of ID and IDA.

## 1. Introduction

Breast milk is considered the optimal nutrition and healthy food for the first six months of life, and breastfeeding is typically complemented with other foods from six months of age until at least 12 months for nearly all infants [1]. A reappraisal of the evidence from a recent expert review for the European Food Safety Authority (EFSA) concluded that for infants across the EU, complementary foods might be introduced safely between the fourth and sixth month [2]. Studies have determined that the iron content in breast milk is low, infants fed with solid food before six months or received iron supplement will decrease the risk of IDA [3]. Anemia is defined as a reduced erythrocyte count or hemoglobin (Hb) value of 5 percentile below the normal hemoglobin value specified for that age in healthy individuals [4]. Therefore, the US Department of Agriculture and the Centers for Disease Control recommend the introduction of complementary food in infants aged between 4 and 6 months [5].

In addition to inadequate iron intake, reduced bioavailability of dietary iron, increased iron requirements, and chronic blood loss is common causes of iron deficiency (ID). ID is currently the most common micronutrient deficiencies worldwide, and a World Health Organization (WHO) survey in 2008 revealed that globally, approximately 293 million (47.4% prevalence) preschool-age children and approximately 56 million (41.8% prevalence) pregnant women suffer from anemia, among which approximately 50% cases are attributable to ID [6]. ID is undoubtedly the major cause of most anemia cases, although other minerals or vitamins can also be responsible for this pathology. Prolonged ID often leads to iron deficiency anemia (IDA) [7]. The American Academy of Pediatrics determined that IDA adversely affects an infant’s social and emotional behavior development and cognitive performance [8]. Approximately 25% of infants in developing countries have IDA, and IDA during pregnancy increases the risk of preterm delivery and adverse perinatal outcomes, such as maternal hemorrhage, sepsis, low birth weight, and possibly poor neonatal health [9,10,11].

Recent studies have indicated that iron content in breast milk is low [12,13,14], and prolonged use of breast milk as the main food in infants might increase the risk of ID and IDA [15]. In Taiwan, the Ministry of Health and Welfare promotes the benefits of breastfeeding for a long period. Although Tsai et al. reported that IDA was associated with prolonged, predominant breastfeeding, the study sample size was small, and no relevant risk factors were evaluated [16]. The purpose of the present study was to investigate the iron status of infants in their first year of life and analyze the relevant influencing factors.

## 2. Participants and Methods

### 2.1. Study Subjects

A cross-sectional study was conducted in three hospitals, namely, Shin Kong Wu Ho-Su Memorial Hospital, Taipei Medical University Hospital, and Shuang Ho Hospital, from October 2012 to January 2014. Overall, 2804 healthy infants aged 1–12 months who came to the well-baby clinic for routine vaccination were screened for eligibility. The inclusion criteria for infants and mothers enrolled were as follows: No systemic diseases (toxemia, hypertension, diabetes mellitus, and heart disease) during pregnancy, age below one year, healthy and without any disease, as determined by a pediatrician. Infants with premature birth, congenital diseases (such as heart, lung, liver, and intestinal diseases), growth disorders, diagnosis of gastrointestinal disorders (nausea, vomiting, pain, flatulence, diarrhea, and malabsorption), and thalassemia were excluded.

The Ethics Committee of Shin Kong Wu Ho-Su Memorial Hospital and Taipei Medical University approved this study in accordance with the International Ethical Guidelines for Biomedical Research Involving Human Subjects and ethical principles of the Declaration of Helsinki (20120901R and 201308004). Written informed consent was obtained from all participants or legal representatives before the study procedures were performed.

### 2.2. Basic Characteristics and Dietary Iron Intake Assessment

The contents of the questionnaire were validated by experts, including the infant’s basic information, such as gestational weeks, chronological age, birth weight, birth length; anthropometrics, such as body length, body weight, and head circumference; and dietary intake. Gestational weeks, chronological age, birth weight, and birth length were recorded per the medical chart. Body length and body weight were measured using an infantometer and weighing scales. The head circumference was measured by applying a plastic tape around the forehead (above the eyebrows) and the occipital protuberance. These measurements were converted to percentiles according to the growth charts for Taiwanese children released by the Ministry of Health and Welfare. The diet was assessed using a semi-quantitative frequency method, and the included food items were tofu, chicken, pork, beef, fish, egg yolk, viscera, vegetables, rice, wheat, baby rice cereal, baby wheat cereal, formula milk, juice, and puree. Complementary food and formula milk intakes were quantified based on standard bowls, spoons, and feeding bottles. Iron intake from the complementary food was calculated using the Nutritional Chamberlain Line, Nutritionist Edition, version 2002 (E-Kitchen Business Corp, Taiwan). Iron intake from formula milk and baby cereal was calculated according to their nutrient labels. Iron content in breast milk was obtained through actual measurement. Breast milk intake was obtained from mothers’ reports. If the mothers were directly breastfeeding their infants, the total volume of breast milk intake was assessed according to the report of Lyu et al. [17]. Pre-pregnancy body weight (kg), gestational weight gain (kg), current weight, and education level were obtained from self-reported data. Current body weight (kg) and height (cm) were measured using an electronic health scale (Tanita corp., Tokyo, Japan). Complete blood count of the mothers, including Hb level, hematocrit (Hct) level, and mean corpuscular volume (MCV) before delivery at 37–40 weeks pregnancy were recorded according to the medical charts.

### 2.3. Breast Milk and Blood Collection

Breast milk from the mothers’ unelated breast was collected using an electric milking machine (Lactina, Medela, Switzerland) for 20 min, which was then mixed, dispensed, and ice-cooled at −80 °C until analysis. Blood samples (4 mL) of infants were collected through an arterial puncture of one arm into vacutainers with and without anticoagulant ethylenediaminetetra-acetic acid (EDTA) by a trained technician. Serum was collected after centrifugation at 1400× *g* for 10 min at 4 °C and immediately sent to Central Laboratory, Shin-Kong Wu Ho-Su Memorial Hospital, for analysis.

### 2.4. Biochemical Analyses

Complete blood count was determined using an automatic blood cell analyzer (Biotecnica Instruments SpA, Roma, Italy). Ferritin was detected using a chemiluminescent immunoassay (Roche Diagnostics, Lewes, UK). Serum iron was analyzed using the ferrozine method (Siemens Healthcare, Marburg, Germany). Total iron binding capacity (TIBC) is the ability of transferrin to bind with iron that was measured by chemistry analyzer using dedicated reagents (Siemens Healthcare, Marburg, Germany). Transferrin saturation (TS, %) represents the percentage of transferrin bound to iron ions, calculated by dividing serum iron concentration by TIBC and multiplying the result by 100.

### 2.5. Breast Milk Iron Content Analysis

Aliquots of 0.5 mL of the mixed breast milk sample was added to 1.5 mL of 70% nitric acid and 0.5 mL of 30% hydrogen peroxide, separately. After mixing and allowing the samples resting for one night, the mixture was digested in a 50 mL polypropylene digestion bottle at 95 °C for 1 h. After cooling at room temperature, the digested sample was diluted using 50 mL of deionized water. Subsequently, 1 mL of the dilution was pipetted into a 15 mL centrifuge tube and diluted with 10 mL of 2% aqueous nitric acid solution to detect the iron content by using inductively coupled plasma mass spectrometry (ICP-MS) (ThermoFisher Scientific, Bremen, Germany) [18]. The iron content in breast milk was calculated using a standard curve constructed using pure iron standards for ICP-MS (Merck, Darmstadt, Germany); with an *R* value of ≥0.99, coefficient of variation (CV) at 2%, and a recovery rate of 80–120%.

### 2.6. Statistical Analysis

The WHO defines ID as a serum ferritin level less than 15.0 ng/mL and IDA as serum ferritin level less than 15.0 ng/mL and Hb less than 10.5 g/dL. We divided the subjects into three groups according to the WHO definitions: The normal, ID, and IDA groups. All data were confirmed to have a normal distribution by using the Kolmogorov–Smirnov test. Data are presented as means ± standard deviations (SDs), median (interquartile range), or percentage. Intergroup differences were determined using one-way ANOVA, followed by the Scheffé method for post hoc test or nonparametric statistics. Pearson’s chi-squared test was used to assess categorical variables. The correlation between Hb and dietary iron intake was determined using the Pearson correlation test. The association between feeding types and anemia was determined using multivariable logistic regression. All data analyses were performed using SPSS (version 19; SPSS Inc., Chicago, IL, USA). Differences were considered significant at *p* < 0.05.

## 3. Results

### 3.1. Participant Characteristics and Infant Anemia Diagnosis

A total of 1368 infants were eligible for this study. However, 779 mothers did not provide consent to extract their infants’ blood, and therefore, the 589 subjects were ultimately enrolled in this study. However, blood draws were unsuccessful in 39 infants. Thus, a total of 550 cases were included for data analysis. Because no introduction of complementary food to infants over six months of age was considered as abnormal feeding, four infants aged eight months and two aged 12 months were excluded accordingly. In addition, 35 infants with a WBC count of >0.000/mm^3^, suspected of infection, were excluded. Data on the iron status analysis of 509 infants are presented in Figure 1.

Anemia was defined according to the WHO criteria, and 49 (10%) and 21 infants (4%) were diagnosed with ID and IDA, respectively, by the physician (Table 1). No statistically significant intergroup differences were noted in terms of sex gestational age, birthweight, body length, body weight, and head circumference of infants. The formula-fed type (52.8%) was the predominant population in the healthy group. The breastfed infants had a significantly higher prevalence of ID (65.3%) and IDA (85.7%) than the mixed-fed and formula-fed infants (*p* < 0.001). No significant intergroup differences were observed in terms of the mother’s age, body mass index (BMI), gestational weight gain, and education.

### 3.2. Analysis of Iron Status

The iron status of the infants and mothers before delivery are presented in Table 2. Biochemical parameters, ferritin, and TS were significantly low in ID and IDA groups. Although the mothers of infants with IDA had a slightly lower Hb level before delivery, no statistical difference was observed.

### 3.3. Iron Intake of Infants

The total iron intake was calculated as the sum of the iron contents in milk and complementary food, as presented in Table 3. Because complementary food was introduced in only 17 infants, and most of the complementary foods were cereals and fruit purees for infants aged 4–6 months, data of the iron intake from complementary food are not shown. During ages 1–6 months, the daily iron intake from milk in the normal, ID, and IDA groups were 3.43, 0.13, and 0.13 mg, respectively. Moreover, the normal group had a higher total iron intake (3.49 mg daily) than the ID (0.13 mg daily) and IDA (0.13 mg daily) groups at ages 1–6 months. The low total iron intake in infants of the ID and IDA groups could be attributed to exclusive breastfeeding, and both ID and IDA were observed in infants aged 4–6 months.

The normal group had the highest iron intake from milk (5.04 mg daily), whereas, the ID and IDA groups had a low iron intake from milk. This could be attributed to the continuous breastfeeding of the infants in the ID and IDA groups. No intergroup differences were noted in terms of the iron intake from complementary food. Therefore, milk was the major source of iron intake in infants aged 7–12 months. Furthermore, we determined that the prevalence of ID and IDA in infants aged 7–12 months was 20.4% and 6.6%, respectively, which was significantly higher than 3.7% and 2.7%, respectively, in infants aged 1–6 months. Our results revealed that infants with ID or IDA before six months were all in aged 4–6 months and breastfed. The medium iron intake in infants aged 4–6 months in the normal, ID and IDA groups were 5.13 mg (6.54), 0.13 mg (0.18), and 0.13 mg (0.29), respectively. In aged 7–12 months, infants with IDA had the highest breastfeeding rate, followed by those with ID.

Iron is essential for hematopoiesis. We observed that the total dietary iron intake was positively correlated with Hb (*r* = 0.292, *p* < 0.001) (Figure 2A). We further divided the data into the following two subgroups: One to six months (Figure 2B) and 7–12 months (Figure 2C). The data revealed that the correlation between total dietary iron intake and Hb in infants aged 7–12 months was higher than in those aged 1–6 months (*r* = 0.390, *p* < 0.001 vs. *r* = 0.130, *p* = 0.028).

### 3.4. Association between Feeding Type and Iron Status

Logistic regression models were employed to identify predictors of ID and IDA. Our data revealed that feeding type was a major indicator in predicting anemia (Table 4). In model 1, no covariates were adjusted, and in model 2, gestational week, birthweight, sex, body weight percentile, body length percentile, and age of infants were adjusted. In model 3, in addition to the aforementioned variables for infants, mothers’ BMI, gestational weight gain, education level, and Hb concentration before delivery were also adjusted. Compared with formula-fed infants, breastfed infants had a higher odds ratio (OR) for ID (OR: 2.715; 95% CI: 1.830–4.030) and IDA (OR: 4.338; 95% CI: 2.108–8.927) in model 1 (*p* < 0.001). Furthermore, ORs for ID and IDA significantly differed after adjustment for variables in models 2 and 3.

## 4. Discussion

The period of infancy constitutes a critical window of growth and brain development, and thus, micronutrient deficiencies during this period may have adverse effects on neurocognitive functions. Iron deficiency is the most prevalent nutritional deficiency among infants in developing countries. Women of childbearing age are at risk of iron deficiency because of poor iron content in the diet, increased demand for iron during pregnancy, and iron loss during menstruation and childbirth. In addition, breastfed infants are vulnerable to developing ID because of rapid growth, depletion of their iron endowment, and low iron content in breast milk and in some complementary foods. In our study, among the 509 mother–infant dyads who submitted complete data, 3.7% and 2.7% of the infants aged below six months had ID and IDA, respectively. However, in infants aged above six months, the prevalence of ID and IDA rapidly increased to 20.4% and 6.6%, respectively. Analysis of the maternal hematologic data revealed that anemia was not detected in the mothers during the routine prenatal examination. The basic characteristics of all infants (Normal, ID, IDA) were similar except for the chronological age and feeding type.

The prevalence of ID and IDA varies greatly among countries worldwide. In India, the prevalence of ID in infants aged 3, 4, and 5 months was 5.4%, 21.4%, and 36.4%, respectively, whereas, that of IDA was 4.6%, 16.7%, and 11.4%, respectively [19]. In Turkey, Germany, and Brazil, the prevalence of ID in infants aged four months were 19.8%, 6%, and 5.7%, the prevalence of IDA in those infants were 9.5%, 0%, and 3.4%, respectively [20,21,22]. The prevalence of ID and IDA in infants before six months of age in our study was concordant with that observed in Germany and Brazil. In Spain, the prevalence of ID and IDA was 9.6% and 4.3%, respectively, in 12-month-old infants [23]. The prevalence of ID was 14.0%, and that of IDA was 9.4% in infants aged 9–12 months in Estonia [24]. In Saudi Arabia, out of the 274 infants aged 6–24 months studied, 126 (51%) were diagnosed as having IDA [25]. Among 619 Korean infants aged 8–15 months old, ID and IDA were diagnosed in 174 (28.1%) and 87 infants (14.0%), respectively [26]. The prevalence of ID and IDA in two cohorts of infants aged nine months old in China was 2.8% and 20.7% and 12.0% and 31.2%, respectively [27]. The prevalence of ID and IDA in infants above six months of age in our study was concordant with that observed in China and Korea.

Breast milk is considered the best food for infants, because it is highly nutritious for infant growth and contains maternal antibodies that provide defense against pathogens. A recent study indicated that human breast milk has very little iron (0.5 mg/L at one month and 0.29 mg/L at 3–5 months) [28]. Our data revealed that the average iron content of breast milk from the mothers of the infants was 0.21 ± 0.06 mg/L. Although iron in human breast milk has higher bioavailability, it may not be sufficient for infants. Therefore, infants’ body iron stores meet most requirements for breast-fed infants during the first six months of life. A study concluded that a normal, healthy, full-term infant has sufficient amounts of iron until approximately 4–6 months of age [29]. Our results revealed that the prevalence of ID and IDA in infants aged 1–6 months was 3.7% and 2.7%, respectively, and these infants were all in aged 4–6 month and breastfed. Hence, breastfeeding for more than four months can slightly increase the risk of ID and IDA with lower iron stores.

Studies have indicated that the risk factors for the development of ID included small-for-gestational-age, infants below 10th percentile weight for gestation, infants of diabetic mothers, very-low-birth weight preterm neonates (VLBW, <1500 g at birth) and infants with lower iron stores [30,31]. Therefore, although the iron status of our subjects seems to be more related to iron intake, studies have shown the smoking, obesity, and childbirth by caesarean section also affected the iron status of infants [32]. These factors were not considered in the present study. In the absence of inflammation, serum ferritin measurement is the most specific test to determine total iron content stored in the body. Our data revealed that infants in ID and IDA groups with lower serum ferritin levels reflected depleted iron stores. Another indicator of iron deficiency was TS. In cases of iron deficiency, serum iron is reduced, and TIBC is increased, resulting in a substantial reduction in TS. A threshold of 16% is generally used to screen for iron deficiency. Our data revealed that subjects with ID and IDA had a TS below the threshold and TS in IDA group was lower than that in ID group.

Another possible reason causing iron depletion in infants below six months of age could be maternal iron deficiency during pregnancy, which results in offspring with inadequate iron storage in the body. Some studies have found little or no correlation between maternal and neonatal iron status, whereas, others have suggested that the fetus was vulnerable to maternal ID [33,34]. Mothers of IDA infants had lower Hb level, although there were no statistical differences. We further analyzed the correction between infants Hb level and mothers Hb level, and the results were not correlated. Although the mother’s iron nutrition status during pregnancy may affect the child’s iron nutrition status, our results revealed that feeding still mainly affects the infant’s iron nutrition status. In addition, the timing of umbilical cord clamping affects iron stores in the newborn. A study indicated that delay in cord clamping increases the red blood cell volume and iron stores in infants [35]. Hence, cord clamping could be a feasible solution to improve the iron status of infants.

The WHO recommends exclusive breastfeeding of infants for the first six months of life. Thereafter, infants should receive complementary foods with continued breastfeeding up to two years of age or above. Taiwan’s health policy also follows this recommendation. In this study, among 328 infants below six months of age, only 17 were introduced to complementary food. Most of these foods were cereals or fruit juices that had low iron content. Thus, in addition to prenatal storage, formula milk contributed more iron to the infants below six months of age. Although human milk has a high bioavailability of iron, the iron content was so low that exclusive breastfeeding increased the risk of ID and IDA. We determined that the total iron intake was positively correlated with Hb level in infants below six months of age, thereby indicating that although prenatal storage was crucial to maintain iron status, additional dietary iron intake can improve the Hb levels and prevent ID and IDA, especially in infants aged 4–6 months. Evidence from randomized control trials suggests that the rate of IDA in breastfed infants could be positively altered by the introduction of solids at four months of age [36]. The American Academy of Pediatrics recommends that exclusively breastfed, full-term infants receive 1 mg/kg of iron supplements per day from the age of four months [37]. Taiwan’s maternal and child policy particularly emphasizes the benefits of breastfeeding. The exclusive breastfeeding rate for infants at six months in Taiwan increased from 20.1% in 2004 to 50.2% in 2011 [38]. Therefore, introducing complementary food early or providing iron supplementation is imperative for infants aged 4–6 months who are exclusively breastfed.

Although infants aged 7–12 months were provided complementary foods, the iron intake from complementary foods was not different among the normal, ID, and IDA groups. Notably, milk was the major source of iron. Upon further analysis of data, we observed that up to 70% of infants with ID and 90% of infants with IDA were still breastfed. This result indicated that infants who are still breastfed for more than six months do not obtain adequate iron from breast milk. This suggests that mothers who still breastfeed after six months could provide high-quality iron-rich foods to reduce the prevalence of ID and IDA. 

When we reported the iron status of infants to their mothers, most of the mothers presented their infants had good growth and not aware of the insufficient iron intake and that it increased the risk of ID or IDA. Conversely, infants who used commercial formulas instead of breast milk had better iron status because most commercial formulas are iron-fortified. Future health policies should educate mothers to prepare high-quality iron-rich foods or provide iron supplementation for breastfed infants.

The maximum iron requirement for infants aged 4–12 months is approximately 1 mg/kg or 10 mg/day. The average full-term infant requires 8 mg of iron daily from approximately six months of age [39]. Our data demonstrated that the median iron intake of infants aged 1–6 months was 3.49 mg/day, and the median iron intake of infants aged 7–12 months was 6.47 mg/day, which could reach the normal iron nutritional status. Infants with IDA had normal growth and no obvious symptoms except for pallor were observed, and no infants were suspected of having anemia by caregivers. Therefore, the presentation of iron deficiency is subtle and can be detected accurately only by medical personnel.

Although our results determined that breastfeeding was associated with an increased the prevalence of ID and/or IDA, the advantages of breast milk are recognized and unquestionable. Thus, more attention should be paid to the problem of iron deficiency in infants, and strategies should be proposed, including improving maternal iron status, introducing high-quality complementary food early, and even providing iron supplement to breastfed infants. The Pediatrics Association of Taiwan revised the guidelines for breastfed infants in 2016. The content includes the following: (1) Encourage full-term infants to start breastfeeding as soon as possible after birth, (2) Continue to breastfeed until one year of age. After one year of age, the mother may continue to breastfeed the infants, (3) Breastfed infants should be started on complementary foods at 4–6 months of age. If no complementary foods are added after four months, oral iron supplementation should be started at 1 mg/kg/day. Iron deficiency affects the development of cognition, nerves, and behavior, some of which are long-term and irreversible [40]. In this study, all infants with ID and IDA improved iron status and anemia after medical iron supplementation and regular checkups by pediatricians.

The strength of the present work is that it is the first to look at the association between breastfeeding and ID and IDA during infancy in Taiwan. The results can be used as a reference for nutrition policymakers. However, the study had some limitations. First, this was a cross-sectional study that was unable to clarify the effects of longitudinal nutritional status on the development of ID and IDA. Second, the sample size was relatively small, and all participants lived in northern Taiwan, which might limit the generalizability of the results. Therefore, additional large-sample, multicenter studies are required. Third, we only prioritized examining several clinical iron markers that are more likely related to iron status. Fourth, the lack of information regarding the initiation of complementary feeding, that could significantly impact iron status in infants. Nevertheless, further studies are warranted, including those for following up infants with iron deficiency in the first year to analyze whether it affects their cognition function and growth, as well as whether early introduction of high-quality iron-rich complementary foods can improve the iron status of infants.

## 5. Conclusions

Our data revealed that exclusive breastfeeding increases the prevalence of ID and IDA in infants aged 4–6 months, and prolonged breastfeed above six months can significantly increase the prevalence of ID and IDA. Although iron in breast milk has good bioavailability, its low iron content results in insufficient iron intake in infants and increases the risk of anemia. Hence, health policies should encourage the early introduction of iron-rich complementary foods and educate mothers to prepare high-quality complementary foods to reduce the risk of anemia, especially in infants above six months. In addition, if no complementary foods are added after four months, oral iron supplementation should be started in breastfeeding infants.

## Figures and Tables

**Figure 1 nutrients-12-00139-f001:**
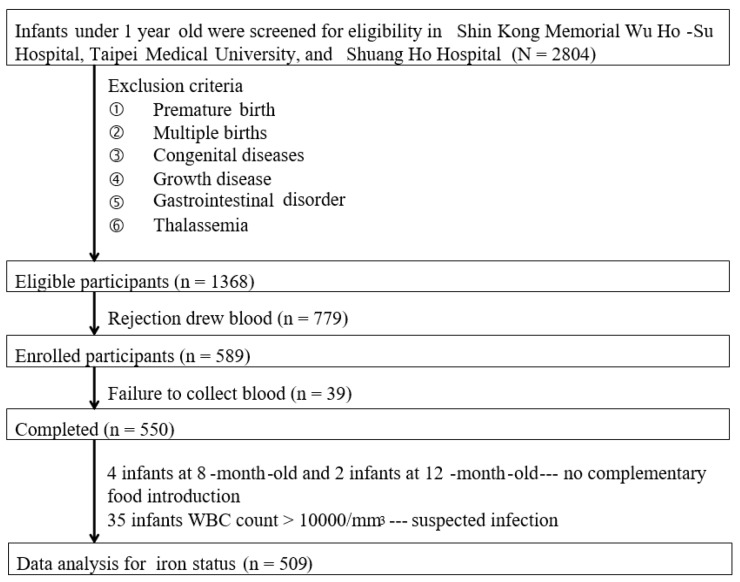
Flowchart of the enrolment of infants.

**Figure 2 nutrients-12-00139-f002:**
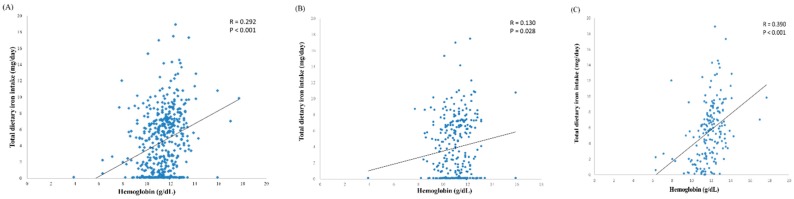
Correlations between total dietary iron intake and hemoglobin at 1–12 months (**A**), 1–6 months (**B**) and 7–12 months (**C**).

**Table 1 nutrients-12-00139-t001:** Demographic characteristics of subjects diagnosed with iron deficiency (ID) or iron deficiency anemia (IDA) in the first year of life ^a^.

Characteristics	Normal *N* = 439	ID ^b^ *N* = 49	IDA *N* = 21	*p* Value
**Infant**				
Number (%)	439 (86)	49 (10)	21 (4)	
Male (%)	236 (53.8)	24 (49.0)	12 (57.1)	0.546
Gestational age	38.4 ± 1.5	38.2 ± 1.6	38.2 ± 1.4	0.673
Chronological age	5.9 ± 4.3	9.9 ± 3.6	9.6 ± 3.4	<0.001 *
Birth weight (kg)	3.0 ± 0.5	3.0 ± 0.6	3.0 ± 0.5	0.919
Body length (percentile)	52.9 ± 30.9	55.8 ± 31.4	37.9 ± 31.3	0.118
Body weight (percentile)	51.2 ± 28.3	51.3 ± 30.0	43.9 ± 30.4	0.568
Head circumference (percentile)	56.4 ± 30.2	50.1 ± 31.2	51.9 ± 20.3	0.424
**Feeding type ^c^**				
Breast-fed (%)	127 (28.9)	32 (65.3)	18 (85.7)	<0.001 *
Mix-fed (%)	80 (18.2)	6 (12.2)	2 (9.5)	
Formula-fed (%)	232 (52.8)	11 (22.4)	1 (4.8)	
**Mother**				
Age (year)	32.6 ± 4.2	32.1 ± 3.6	32.4 ± 4.7	0.791
BMI (kg/m^2^)	21.3 ± 3.7	20.8 ± 5.5	20.7 ± 3.8	0.654
Gestational weight gain (kg)	14.3 ± 7.9	16.3 ± 11.9	11.6 ± 5.5	0.125
Education (year)	14.6 ± 2.3	15.0 ± 2.3	15.2 ± 2.0	0.382

^a^ Values are expressed as mean ± SD or *n* (%); ^b^ According to definitions by the World Health Organization, iron deficiency (ID): Serum ferritin < 15.0 ng/mL, iron deficiency anemia (IDA): Serum ferritin < 15.0 ng/mL and hemoglobin < 10.5 g/dL.; ^c^ Definition of feeding type: Breast-fed means that the dairy products in the infant’s diet are all breast milk; Mix-fed means that the dairy products in the infant’s diet include breast milk and formula milk; Formula-fed refers to the dairy products in the infant’s diet only formula milk. * Differences between groups were tested using one-way ANOVA, followed by the Scheffé method for a post hoc or Chi-square test; *p* < 0.05 was considered statistically significant.

**Table 2 nutrients-12-00139-t002:** Hematologic data of subjects ^a^.

Classification	Normal *N* = 439	ID ^b^ *N* = 49	IDA *N* = 21	*p* Value ^c^
**Infant**				
Hb (g/L)	11.4 ± 1.3 ^a^	11.6 ± 0.7 ^a^	9.2 ± 1.4 ^b^	<0.001
Ferritin (ng/mL)	55.0 (98.4) ^a^	10.2 (5.7) ^b^	5.2 (5.7) ^b^	<0.001
TS (%)	22.5 ± 11.3 ^a^	11.5 ± 5.5 ^bc^	7.0 ± 4.5 ^c^	<0.001
**Maternal (before delivery)**				
Hb (g/L)	12.0 ± 1.9	12.1 ± 1.4	11.3 ± 1.4	0.361
Hct (%)	36.2 ± 4.6	36.6 ± 3.2	34.5 ± 3.7	0.333
MCV (fl)	86.8 ± 8.2	87.1 ± 8.4	86.7 ± 8.1	0.584

^a^ Data are presented as mean ± SD or median (interquartile range); ^b^ ID, iron deficiency; IDA, iron deficiency anemia; Hb, hemoglobin; TS, transferrin saturation; Hct, hematocrit; MCV, mean corpuscular volume; ^c^ Means in the column with different superscripts indicate a significant difference (*p* < 0.05), tested using one-way ANOVA, followed by the Scheffé method for post hoc test or the Kruskal–Wallis test.

**Table 3 nutrients-12-00139-t003:** Iron intake of infants aged 1–6 months and 7–12 months ^a^.

Parameters	Normal	ID ^b^	IDA	*p* Value
**1–6 months**				
Number (%)	307 (93.6)	12 (3.7)	9 (2.7)	
Chronological age	3.4 ± 1.8	5.0 ± 1.2	4.8 ± 1.1	0.007 *
Breastfeed (%)	174 (56.7)	12(100)	9 (100)	0.001 *
Iron intake from milk (mg/day) ^c^	3.43 (6.62)	0.13 (0.03)	0.13 (0.17)	0.003 *
Total iron intake (mg/day) ^d^	3.49 (6.81)	0.13(0.18)	0.13(0.17)	0.007 *
**7–12 months**				
Number (%)	132 (73.0)	37 (20.4)	12 (6.6)	
Chronological age	11.3 ± 2.9	11.2 ± 2.9	11.3 ± 2.0	0.976
Breastfeed (%)	21 (15.9)	26 (70.3)	11 (91.7)	<0.001 *
Iron intake from milk (mg/day) ^c^	5.04 (3.92)	0.14 (2.01)	0.14 (0.01)	<0.001 *
Iron intake from complementary food (mg/day)	1.04 (1.00)	1.29 (2.00)	1.20 (2.00)	0.840
Total iron intake (mg/day)	6.47 (4.35)	2.28 (3.02)	1.33 (0.98)	<0.001 *

^a^ Data are presented as median (interquartile range); ^b^ ID, iron deficiency; IDA, iron deficiency anemia; ^c^ Iron intake from milk was the sum of iron intake from formula milk and breast milk.; ^d^ Total iron intake was the sum of iron intake from milk and complementary food; * Intergroup differences were tested using the Kruskal–Wallis test or chi square test; *p* < 0.05 was considered statistically significant.

**Table 4 nutrients-12-00139-t004:** Association between feeding type and iron status.

Variables	β	SE ^a^	OR	95% CI	*p* Value
**Model 1 ^b^**					
ID	0.999	0.201	2.715	1.830–4.030	<0.001 *
IDA	1.467	0.368	4.338	2.108–8.927	<0.001 *
**Model 2**					
ID	0.984	0.201	2.674	1.803–3.966	<0.001 *
IDA	1.361	0.369	3.901	1.893–8.042	<0.001 *
**Model 3**					
ID	0.769	0.232	2.157	1.369–3.399	0.001 *
IDA	1.434	0.437	4.196	1.780–9.887	0.001 *

^a^ SE, standard error of mean; OR, odds ratio; 95% CI, 95% confidence interval; ^b^ Model 1: Not adjusted; Model 2: Adjusted for gestational week, birthweight, sex, body weight percentile, body length percentile, age; Model 3: Adjusted for gestational week, birthweight, sex, body weight percentile, body length percentile, age of infants, as well as mothers’ BMI, gestational weight gain, education level, and hemoglobin level before delivery; * *p* < 0.05.

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
