# Peer review of "Iron Status of Infants in the First Year of Life in Northern Taiwan"

_nutrients, 2020, doi:10.3390/nu12010139_

Round 1

Reviewer 1 Report

Nutrients

Manuscript ID: nutrients-671956

Title: Iron status of infants in the first year of life in Northern Taiwan

Authors: Chiao-Ming Chen, Shu-Ci Mu, Chun-Kuang Shih, Yi-Ling Chen, Li-Yi Tsai, Yung-Ting Kuo, In-Mei Cheong, Mei-Ling Chang, Yi-Chun Chen, Sing-Chung Li

The authors have investigated the prevalence of ID and IDA, and the association between feeding type and iron nutritional status in 509 infants in northern Taiwan. The results revealed that 49 (10%) and 21 (4%) 35 infants in their first year of life had ID and IDA and that this was due to exclusive prolonged breastfeeding. They conclude that iron-rich complementary foods should be introduced at the age of 4–6 months.

The study includes a large number of infants, it is very well-written and interesting.

A major point is missing data on age at examination:

Line 118: Iron content in breast milk was obtained through actual measurement. In table 3 data are given for age 1-6 month: was infants also examined at this age?

Chronological age of the infants should be included in table 1.

Age at examination might influence ironstores and period of complementary feeding.

Line 130: was arterial puncture performed, not venous?

Table 2: Ferritin in normal infants had a very large SD (100). Maybe it would be better to use median and interquartile range?

Author Response

Response to Reviewer 1 Comments

Point 1: Line 118: Iron content in breast milk was obtained through actual measurement. In table 3 data are given for age 1-6 month: was infants also examined at this age?

Response 1: Thank you for your pointing out. Both breast milk and infant’s blood draw were collected on the same day. Breast milk was frozen at -80℃ and determined iron content in breast milk by ICP-MS. The infant's blood is immediately sent to the laboratory for biochemical analysis. Yes, all infants were examined at their chronological age.

Point 2: Chronological age of the infants should be included in table 1.

Response 2: Thank you for your valid and constructive comments. The chronological age of the infants was included in table 1.

Point 3: Age at examination might influence iron stores and period of complementary feeding.

Response 3: We agree with your opinion on this point. Our results revealed that infants with ID or IDA before 6 months were all in aged 4-6 months and breastfed, it may be related to insufficient iron storage. But, our data also showed that enough iron from diet (normal group) could avoid the occurrence of ID and IDA. The chronological age in each group was approximately 11 months in infants aged 7-12 months. We only have a cross-sectional survey of dietary iron intake and didn’t investigate the period of complementary feeding. Therefore, it is our limitation to evaluate iron status in period of complementary feeding unless longitudinal study used in future.

Point 4: Line 130: was arterial puncture performed, not venous?

Response 4: This study was part of a survey “The effects of breastfeeding on iron and vitamin D status in infants.” A total of 4 cc blood is needed for all biochemical tests. At the beginning, we started drawing blood by venous, but the blood flow was slow and coagulated easily, resulting in insufficient blood volume for analysis. So we switched to arterial puncture and it was easier to obtain sufficient blood volume. All infants were well cared by medical staff without comorbidities after the blood was drawn.

Point 5: Table 2: Ferritin in normal infants had a very large SD (100). Maybe it would be better to use median and interquartile range?

Response 5: We agree with your opinion on this point. The updated concentration of ferritin was showed in table 2.

Reviewer 2 Report

The association between breastfeeding and iron deficiency during deficiency is increasingly studied. This current manuscript seeks to establish association between breastfeeding and iron intake with iron status during infancy in northern Taiwan. This is an interesting paper, but some areas could be improved by being more precise, particularly on the age of the infants in each group considered. Please see below.

Abstract

L33: Please specify the age range

L38-39: it would be more meaningful to compare biomarkers of iron status between breastfed infants and formula/mixed fed infants

L45-46: presenting data adjusted for all confounding factors (including maternal Hb levels and infant age, i.e. model 3) would be more relevant.

L48: delete ‘at the age of’

L49: replace ‘among those who are still’ by ‘when’

Introduction

L78-81: Please be more precise on what you mean by ‘long time’ and  ‘prolonged’

L80: remove ‘no studies have yet investigated the iron status of infants’ as this is not correct

Material and methods:

L92: please spcify if there was a minimum age or of this was from birth

L124-125: Please specify at what stage of pregnancy were Hb, MCV, Hct measured in mothers

L130: Why was arterial blood taken as opposed to less invasive venous or capillary blood taking?

L161-62: Correlations between infant iron status and maternal iron status (Hb) should be performed too

L190: please amend the legend so that * refers to chi square (association) only rather than one way anova (valid for all other data)

Results

L173-174: Figure 1 should be referred to as the flowchart of the enrolment of infants rather than ‘data on iron status’

L175-176: add breakdown of numbers for all exclusion criteria (premature birth, n = …, etc)

L185-186: Table 1 should include the infant’s age too (Iron status is mostly determined by iron stores at birth, and children nearer 6Mo may be more deficient than those aged 2 or 4 Mo)

L197: In table 2, check the superscript for TS in the IDA group, is it b (different from Normal) or c (different from both normal and ID)?

L220: Table 3 should show average infant age for each sub category and breastfeeding data

L227-31: it would be interesting to know if the correlation disappears in younger infants (e.g. < 4Mo)

Discussion

L265: the last sentence is unclear. Do you mean similar between ID and IDA compared with normal iron status? Did infant age differ between the groups?

L269: please specify the infant age for the Saudi study too

L275-76: please specify whether the data are consistent with the literature for infants < 6Mo too, if data is available.

L281: please add standard deviation or another index of variability

L286-287: the fact that all ID and IDA are > 4Mo and exclusively breastfed should appear in the results section (Table 2 shows formula fed and mix fed infants also had ID or IDA, were they all aged > 6 Mo?)

L288: please specify in infants with lower iron stores

L294-95: Please rephrase as ferritin levels < 15ug/L were what defined ID and IDA

L307-310: The part on the possibility of iron deficiency without anemia in the mother is a bit unclear and also needs to be rephrased.

312-313: Please replace ‘this finding’ by ‘cord clamping’, or ‘this method’ could greatly improve…

L324-325: The age range 4-6 Mo should have been highlighted in the results section

L338: Please change to: ‘This suggests that mothers who still breastfeed after 6Mo could …’

L339: it is unclear how the data revealed that mothers were unaware of the insufficient iron intake

L346-47: the median iron intake in infants aged 4-6Mo (rather than 1-6Mo) would be more relevant

L349-50: the results did not indicate that as the difference in weight and length was not significant

L353: please delete ‘hence’

L354-55: it would be more accurate to state that breastfeeding was associated with an increased prevalence of ID and/or IDA.

L357: please delete ‘the’

L358: please add ‘to breastfed infants’

L366: please define ‘recovered well’

L368-69: please rephrase, as ID and IDA are defined by low iron status (e.g. first to look at association between breastfeeding and ID and IDA during infancy in Taiwan?)

Conclusion

L387: Please replace nonetheless with ‘in addition’

Author Response

Response to Reviewer 2 Comments

Abstract

L33: Please specify the age range

Response: The age range (1-12 months) was specified in L33

L38-39: it would be more meaningful to compare biomarkers of iron status between breastfed infants and formula/mixed fed infants

Response: We appreciate your keen comments. Our purpose is to understand the iron nutritional status of infants and to explore the possible reason of iron deficienc in the first year of life in Northern Taiwan, rather than understanding iron nutrition in infants with different feeding styles. Therefore, we have not grouped and discussed according the opinions of reviewers.

L45-46: presenting data adjusted for all confounding factors (including maternal Hb levels and infant age, i.e. model 3) would be more relevant.

Response: Thank you for your pointing out. In L44-48, the setence was updated ‘the odds ratio for ID was 2.157 (95% confidence interval [CI]: 1.369–3.399) and that for IDA was 4.196 (95% CI: 1.780–9.887) among breastfed infants (p < 0.001) after adjusted for all confounding factors (including gestational week, birthweight, sex, body weight percentile, body length percentile, age of infants, mothers’ BMI, gestational weight gain, education level, and hemoglobin level before delivery)’

L48: delete ‘at the age of’

Response: In L48, the setence was updated.

L49: replace ‘among those who are still’ by ‘when’

Response: In L49, the setence was updated.

Introduction

L78-81: Please be more precise on what you mean by ‘long time’ and  ‘prolonged’

Response: In L82, the setence was updated ‘the Ministry of Health and Welfare promotes the benefits of breastfeeding for a long period’

L80: remove ‘no studies have yet investigated the iron status of infants’ as this is not correct

Response: In L81, the setence was removed.

Material and methods:

L92: please spcify if there was a minimum age or of this was from birth

Response: In L91-92, the setence was updated ‘Overall, 2804 healthy infants aged 1-12 months who came to the well-baby clinic for routine vaccination were screened for eligibility’

L124-125: Please specify at what stage of pregnancy were Hb, MCV, Hct measured in mothers

Response: In L125-128, Complete blood count of the mothers, including Hb level, hematocrit (Hct) level, and mean corpuscular volume (MCV) before delivery at 37-40 weeks pregnancy were recorded according to the medical charts.

L130: Why was arterial blood taken as opposed to less invasive venous or capillary blood taking?

Response: This study is a part of a breastfeeding program for iron and vitamin D status of infants in Northern Taiwan. A total of 4 cc of blood is needed for all biochemical tests. At the beginning, we started with venous blood, due to the blood collection needle is thin, the blood flow is slow and it is easy to clot. In addition, the blood volume is not enough for analysis, so later arterial puncture is adopted, and it is easier to obtain sufficient blood volume. All infants were well cared by medical staff without comorbidities after the blood was drawn.

L161-62: Correlations between infant iron status and maternal iron status (Hb) should be performed too

Response: Thank you for your pointing out. We have done this part. Maybe our infants were not newborns, there was no correlation between infant iron status and maternal iron status (Hb). Postnatal feeding also affected infant iron nutritional status. Our results showed that feeding type was the major factor affecting iron status.

L190: please amend the legend so that * refers to chi square (association) only rather than one way anova (valid for all other data)

Response: We added chronological age in table 1. The chronological age is also statistically different, so the original representation is maintained in L196.

Results

L173-174: Figure 1 should be referred to as the flowchart of the enrolment of infants rather than ‘data on iron status’

Response: In L179, the setence was updated ‘Flowchart of the enrolment of infants’ in Figure 1.

L175-176: add breakdown of numbers for all exclusion criteria (premature birth, n = …, etc)

Response: Thank you for your pointing out. At the time of screening, medical staff only recorded the total number of cases excluded and missed to record the numbers of each ineligible cases. We are sorry can not provid this information.

L185-186: Table 1 should include the infant’s age too (Iron status is mostly determined by iron stores at birth, and children nearer 6Mo may be more deficient than those aged 2 or 4 Mo)

Response: The chronological age of the infants was included in table 1.

L197: In table 2, check the superscript for TS in the IDA group, is it b (different from Normal) or c (different from both normal and ID)?

Response: Thank you for your pointing out. The superscript for TS in the ID and IDA were updated in Table 2.

L220: Table 3 should show average infant age for each sub category and breastfeeding data

Response: Both chronological age and breastfeed (%) were showed in Table 3.

L227-31: it would be interesting to know if the correlation disappears in younger infants (e.g. < 4Mo)

Response: There are 225 infants in 1-4 months. Infants’ serum Hb and total dietary iron intake was positive correlation, but no significant. R=0.223, p=0.084.

Discussion

L265: the last sentence is unclear. Do you mean similar between ID and IDA compared with normal iron status? Did infant age differ between the groups?

Response: In L274-276, the setence was updated ‘the basic characteristics of all infants (Normal, ID, IDA) were similar except for the chronological age and feeding type’

L269: please specify the infant age for the Saudi study too

Response: In L277, the infant age for the India, Saudi was specified.

L275-76: please specify whether the data are consistent with the literature for infants < 6Mo too, if data is available.

Response: In L281-283, the prevalence of ID and IDA in infants before 6 months of age in our study was concordant with that observed in Germany and Brazil.

L281: please add standard deviation or another index of variability

Response: In L294-295, breast milk from the mothers of the infants was 0.21 ± 0.06 mg/L.

L286-287: the fact that all ID and IDA are > 4Mo and exclusively breastfed should appear in the results section (Table 2 shows formula fed and mix fed infants also had ID or IDA, were they all aged > 6 Mo?)

Response: In L300-301, these infants were all in aged 4–6 months and breastfed.

Yes, the Table 1 shows formula fed and mix fed infants also had ID and IDA, and all of them were aged > 6 months.

L288: please specify in infants with lower iron stores

Response: In L305, add’ with lower iron store’

L294-95: Please rephrase as ferritin levels < 15ug/L were what defined ID and IDA

Response: In L310-311, the setence was updated ‘Our data revealed that infants in ID and IDA groups with a lower serum ferritin levels reflected depleted iron stores’

L307-310: The part on the possibility of iron deficiency without anemia in the mother is a bit unclear and also needs to be rephrased.

Response: In L320-324, the setence was updated ‘Mothers of IDA infants had lower Hb level, although there were no statistical differences. We further analyzed the correction between infants Hb level and mothers Hb level, and the results were not correlated. Although the mother's iron nutrition status during pregnancy may affect the child's iron nutrition status, our results revealed that feeding still mainly affects the infant's iron nutrition status’

312-313: Please replace ‘this finding’ by ‘cord clamping’, or ‘this method’ could greatly improve…

Response: In L326-327, the setence was updated ‘Hence, cord clamping could be a feasible solution to improve the iron status of infants’

L324-325: The age range 4-6 Mo should have been highlighted in the results section

Response: In L225-229, the setence was highlighted ‘Our results revealed that infants with ID or IDA before 6 months were all in aged 4-6 months and breastfed’

L338: Please change to: ‘This suggests that mothers who still breastfeed after 6Mo could …’

Response: In L352, the setence was updated ‘This suggests that mothers who still breastfeed after 6 months could…’

L339: it is unclear how the data revealed that mothers were unaware of the insufficient iron intake

Response: In L354, the setence was updated ‘When we reported the iron status of infants to their mothers, most of mothers presented their infants had good growth and not aware of the insufficient iron intake and that it increased the risk of ID or IDA.’

L346-47: the median iron intake in infants aged 4-6Mo (rather than 1-6Mo) would be more relevant

Response: In L225-229, the medium iron intake in infants aged 4-6 months in the normal, ID and IDA groups were 5.13 mg (6.54), 0.13 mg (0.18), and 0.13 mg (0.29), respectively.

L349-50: the results did not indicate that as the difference in weight and length was not significant

Response: In L364-366, the setence was updated ‘Infants with IDA had normal growth and no obvious symptoms except for pallor were observed, and no infants were suspected of having anemia by caregivers.’

L353: please delete ‘hence’

Response: In L366, ‘hence’ was deleted.

L354-55: it would be more accurate to state that breastfeeding was associated with an increased prevalence of ID and/or IDA.

Response: In L368-369, the setence was updated ‘Although our results determined that breastfeeding was associated with an increased the prevalence of ID and/or IDA’

L357: please delete ‘the’

Response: In L371, ‘the’ was deleted.

L358: please add ‘to breastfed infants’

Response: In L372, the ‘breastfed’ was added.

L366: please define ‘recovered well’

Response: In L380, the setence was updated ‘improved iron status and anemia’

L368-69: please rephrase, as ID and IDA are defined by low iron status (e.g. first to look at association between breastfeeding and ID and IDA during infancy in Taiwan?)

Response: In L382, the setence was updated ‘it is the first to look at the association between breastfeeding and ID and IDA during infancy in Taiwan…’

Conclusion

L387: Please replace nonetheless with ‘in addition’

Response: In L401, the ‘nevertheless’ was replaced ‘in addition’

Reviewer 3 Report

This manuscript reports on the prevalence of iron deficiency and iron deficiency anaemia in infants from Taiwan. While this is a cross-sectional study, it does provide important information on iron intakes and status of infants in this population.

Abstract - the layout of the results section in the abstract is a little confusing and slightly repetitive. This should be shortened and more focused on the main findings of the paper.

Was a measure of inflammation measured in this cohort? Ferritin is an acute-phase reactant so inflammation is also measured to validate the ferritin results.

Figure 1 - Almost 1500 infants weren't eligible for this study so it would be helpful to see the breakdown of numbers that were excluded during the screening process, not just the list of reasons.

The paper has given no indication of the mean or median age of the infants included in this study. This information should be added to Table 1.

What was the median or mean age of beginning complementary feeding in this population? This is an important consideration when looking at iron status indices. Could this information also be added to Table 1?

In Table 1, the feeding method of infants is provided. The authors should clarify their definitions of these methods, particularly for breast-fed - is this exclusively breast-fed as per the WHO definition (i.e. no food) or how has this study defined it? There are so many different definitions of feeding used across the world so it is important to provide clarity for the reader on this.

Maternal iron supplementation use during pregnancy should be added into Table 1, not just outlined in the text.

Table 3 - a breakdown of iron from both breast milk and infant formula would be more helpful than "iron intake from milk". As this study is in infants up to 12 months, their feeding methods will change with time so this should be reflected in the iron intake data too.

Discussion line 289 - it is now acknowledged that there are a number of other maternal and pregnancy-related factors that can influence neonatal iron status at birth, aside from those that the authors have mentioned. These include maternal smoking, obesity and delivery by Caesarean section. It is worth considering that iron intake is not the only contributor to iron status in full-term infants as even term infants can be born with low iron stores which have been shown to track throughout infancy and early childhood.

Author Response

Response to Reviewer 3 Comments

Point 1: Abstract - the layout of the results section in the abstract is a little confusing and slightly repetitive. This should be shortened and more focused on the main findings of the paper.

Response 1: Thank you for your valid and constructive comments. We had shortened and focused the main findings of our manuscript in abstract.

Point 2: Was a measure of inflammation measured in this cohort? Ferritin is an acute-phase reactant so inflammation is also measured to validate the ferritin results.

Response 2: Our authors agree with your keen opinion that ferritin could increase due to infection or inflammation. We did not measure CRP to exclude inflamed or infected infants due to limited blood volume. In addition, infants were clinical checked by a pediatrican and the white blood cells above 10000 mm3 in counts are excluded by the pediatrician.

Point 3:  Figure 1 - Almost 1500 infants weren't eligible for this study so it would be helpful to see the breakdown of numbers that were excluded during the screening process, not just the list of reasons.

Response 3: At the time of screening, medical staff only recorded the total number of cases excluded and missed to record the numbers of each ineligible cases. We are sorry can not provid this information.

Point 4: The paper has given no indication of the mean or median age of the infants included in this study. This information should be added to Table 1.

Response 4: We appreciate your valid and constructive comments. The chronological age of the infants was included in table 1.

Point 5: What was the median or mean age of beginning complementary feeding in this population? This is an important consideration when looking at iron status indices. Could this information also be added to Table 1?

Response 5: Thank you for your pointing out. We have added the chronological age in table 1. We only have a cross-sectional survey of dietary iron intake from milk and complementary food and did not investigate when to introduce complementary foods. Therefore, it is our limitation to evaluate iron status in period of complementary feeding unless longitudinal study used in future. In our experience, pediatricians will remind mothers to introduce complementary foods during a health checkup in infant aged 6 months. Almost all infants over 6 months in our study started complementary feeding. There are only 6 infants without complementary feeding and considered abnormally fed and were excluded.

Point 6: In Table 1, the feeding method of infants is provided. The authors should clarify their definitions of these methods, particularly for breast-fed - is this exclusively breast-fed as per the WHO definition (i.e. no food) or how has this study defined it? There are so many different definitions of feeding used across the world so it is important to provide clarity for the reader on this.

Response 6: We agree with your opinion on this point. We specified the feeding type in Table 1 (L194). Definition of feeding type: Breast-fed means that the dairy products in the infant's diet are all breast milk; Mix-fed means that the dairy products in the infant's diet include breast milk and formula milk; Formula-fed refers to the dairy products in the infant's diet only formula milk. The exclusively breast-fed is defined as only breast milk in infant diet. Therefore, we cannot claim that our subjects are exclusively breast-fed due to a small number of infants below 6 months who were breastfed also had complementary feeding. However, all above 6 month infants used complementary feeding.

Point 7: Maternal iron supplementation use during pregnancy should be added into Table 1, not just outlined in the text.

Response 7: Thank you for your valid and constructive comments. Due to the differences in the form and dosage of iron supplements, the time of supplementation was also different (pregnancy 1st, 2nd, and 3rd trimester), it is difficult to show iron supplement status of mothers in this manuscript. In the future, we will further analze the correlation between iron supplementation and iron status during pregnancy. In order to focus on the main finding, we delete this part of discussion in the manuscript.

Point 8: Table 3 - a breakdown of iron from both breast milk and infant formula would be more helpful than ‘iron intake from milk’. As this study is in infants up to 12 months, their feeding methods will change with time so this should be reflected in the iron intake data too.

Response 8: Thank you for your pointing out. Our data revealed total iron intake was positive related with Hb of infants. Duo to the iron content in breast milk was low, dividing into iron from breast milk and formula was pointless. But we added the breastfed rate in each groups, and it showed a higher breastfeeding rate in ID and IDA groups. We only have a cross-sectional survey of dietary iron intake from milk and complementary food and did not investigate the feeding methods changes with time by longitudinal.

Point 9: Discussion line 289 - it is now acknowledged that there are a number of other maternal and pregnancy-related factors that can influence neonatal iron status at birth, aside from those that the authors have mentioned. These include maternal smoking, obesity and delivery by Caesarean section. It is worth considering that iron intake is not the only contributor to iron status in full-term infants as even term infants can be born with low iron stores which have been shown to track throughout infancy and early childhood.

Response 9: We appreciate your valid and constructive comments. In L305, the setence was updated ’Therefore, although the iron status of our subjects seems to be more related to iron intake, studies have shown the smoking, obesity, and childbirth by caesarean section also affected the iron status of infants [32]’. These factors was not considered in present study’

Round 2

Reviewer 3 Report

The majority of my original suggestions have been addressed by the authors. Although, the lack of information regarding the initiation of complementary feeding, which can significantly impact iron status in infants, remains a limitation of this study and the authors should acknowledge this further in their discussion. A further small suggestion regarding the wording in the discussion and abstract would be to refer to your findings as prolonged exclusive breastfeeding, as this appears to be associated with poorer iron status rather than just breastfeeding itself, as we know breastfeeding remains the gold standard feeding method for infants.

Author Response

Point: The majority of my original suggestions have been addressed by the authors. Although, the lack of information regarding the initiation of complementary feeding, which can significantly impact iron status in infants, remains a limitation of this study and the authors should acknowledge this further in their discussion. A further small suggestion regarding the wording in the discussion and abstract would be to refer to your findings as prolonged exclusive breastfeeding, as this appears to be associated with poorer iron status rather than just breastfeeding itself, as we know breastfeeding remains the gold standard feeding method for infants.

Response: Thank you for your valid and constructive comments. We add a limitation in L393-394, 'Fourth, the lack of information regarding the initiation of complementary feeding, this could significantly impact iron status in infants'

All infants over 6 months in our study were provided complementary foods. We also excluded 6 infants without complementary feeding in our study. In addition, we can only say that prolonged breastfeeding increased the prevalence of ID and IDA, we cannot say that prolonged exclusively breastfeeding increased the prevalence of those. Respecting your opinion, both sentences were updated in abstract (L51) and conclusion (L400)